# The ConceptARC Benchmark: Evaluating Understanding and Generalization in the ARC Domain

**Arseny Moskvichev** *arseny.moskvichev@gmail.com*
*Santa Fe Institute*

**Victor Vikram Odouard** *vicviod@gmail.com*
*Santa Fe Institute*

**Melanie Mitchell** *mm@santafe.edu*
*Santa Fe institute*

**Reviewed on OpenReview:** *https://openreview.net/forum?id=8ykyGbtt2q*

## Abstract

The abilities to form and abstract concepts are key to human intelligence, but such abilities remain lacking in state-of-the-art AI systems. There has been substantial research on conceptual abstraction in AI, particularly using idealized domains such as Raven's Progressive Matrices and Bongard problems, but even when AI systems succeed on such problems, the systems are rarely evaluated in depth to see if they have actually grasped the concepts they are meant to capture.

In this paper we describe an in-depth evaluation benchmark for the Abstraction and Reasoning Corpus (ARC), a collection of few-shot abstraction and analogy problems developed by Chollet (2019). In particular, we describe ConceptARC, a new, publicly available benchmark in the ARC domain that systematically assesses abstraction and generalization abilities on a number of basic spatial and semantic concepts. ConceptARC differs from the original ARC dataset in that it is specifically organized around "concept groups"—sets of problems that focus on specific concepts and that vary in complexity and level of abstraction. We report results on testing humans on this benchmark as well as three machine solvers: the top two programs from a 2021 ARC competition and OpenAI's GPT-4. Our results show that humans substantially outperform the machine solvers on this benchmark, showing abilities to abstract and generalize concepts that are not yet captured by AI systems. We believe that this benchmark will spur improvements in the development of AI systems for conceptual abstraction and in the effective evaluation of such systems.

## 1 Introduction

Forming and abstracting concepts is at the heart of human intelligence (Carey, 2011; Hofstadter, 1995; Lake et al., 2017). These abilities enable humans to understand and create internal models of the world, to use these models to make sense of new information, often via analogy, and to decide how to behave in novel situations. Giving machines such abilities was one of the key goals of McCarthy et al.'s 1955 Dartmouth AI workshop, but these are precisely the capabilities that are still largely lacking in today's AI systems (Mitchell, 2021; Mitchell and Krakauer, 2023).

In AI, research on concept formation and abstraction often utilizes idealized domains that capture some of the essential aspects of abstraction and analogy in the real world. In such domains one can be explicit about the assumed prior knowledge without requiring the open-ended knowledge involved in real-world language and imagery. Examples of idealized domains that require abstraction and analogy abilities include Raven's Progressive Matrices (Carpenter et al., 1990; Zhang et al., 2019), Copycat letter-string analogies (Hofstadter

and Mitchell, 1994), Bongard problems (Bongard, 1970; Foundalis, 2023), and the Abstraction and Reasoning Corpus (ARC) (Chollet, 2019). The latter three especially, which require the solver to *generate* answers to problems (rather than selecting from candidate answers), remain open challenges for AI systems.

There has been substantial research on developing AI systems to solve problems in each of these domains (Małkiński and Mańdziuk, 2022b). Symbolic AI systems have been developed to solve many of the original Raven's problems (Lovett and Forbus, 2017) and deep learning methods have surpassed human accuracy on automatically generated Raven's-like problems (Małkiński and Mańdziuk, 2022a). Particular subsets of Copycat letter-string problems have been solved by early "active symbol" methods (Hofstadter and Mitchell, 1994) and more recently by large language models (Webb et al., 2022). Simple Bongard problems have been recently addressed by program induction methods (Sonwane et al., 2021), and automatically generated Bongard-like problems have been tackled by deep learning systems (Nie et al., 2020). ARC problems were the subject of a 2020 Kaggle challenge (Kaggle.com, 2020) and a limited number were solved by program-synthesis approaches (Alford et al., 2022; Banburski et al., 2020; de Miquel Bleier, 2020; Wind, 2020a). However, few of these efforts have probed the extent to which AI systems have actually grasped the abstract concepts that these various problems are meant to capture. More specifically, if an AI system is able to solve a problem involving a specific concept, to what extent will it be able to solve other problems that target the same concept, including problems that instantiate the concept in a quite different manner? Such generalization abilities would be crucial to any AI system operating in the real world.

In this paper, we examine how to evaluate the degree to which an AI system has learned or understood a concept in a generalizable way. Machine learning systems are typically developed by randomly splitting a set of examples into training and test sets. However, this kind of evaluation does not systematically test for the kind of learning and understanding that is needed for "out of distribution" generalization. Indeed, it has been shown many times that machine learning systems can learn "shortcuts" that produce high accuracy on the test set but that do not generalize more broadly (Geirhos et al., 2020; Lapuschkin et al., 2019). To evaluate AI systems, in particular systems that are claimed to perform abstract reasoning, new evaluation methods and benchmarks are needed that specifically test that the system has grasped the relevant abstract concepts.

We propose a systematic *concept-based* evaluation method, in which test examples are designed to instantiate variations on chosen concepts. If a system performs well on a range of such examples that vary in complexity and degree of abstraction, that performance provides strong evidence that the system has understood the concept in a generalizable way. In previous work we applied such an evaluation method to programs that exceeded human accuracy on the RAVEN corpus (Odouard and Mitchell, 2022). Our evaluation provided evidence that, while attaining high accuracy on the test set, these programs had not actually learned generalizable abstract concepts. In this paper we propose a concept-based evaluation benchmark for the ARC domain. We discuss why ARC is an excellent domain for studying concept formation and abstraction in both humans and AI systems, but we argue that the original ARC test examples do not systematically evaluate concept understanding.

Our contributions in this paper are (1) the creation of a new concept-based evaluation benchmark for the ARC domain and (2) results from our studies using this benchmark to evaluate state-of-the-art programs that solve ARC problems, as well as human performance on this benchmark. Our results show that humans exhibit strong conceptual generalization abilities in the ARC domain, as compared with much weaker abilities in current AI programs, both those designed for this domain and more general-purpose large language models. We believe that our benchmark, and future extensions of it, will spur improvements in the development of AI systems for conceptual abstraction and in the effective evaluation of such systems.

## 2   The Abstraction and Reasoning Corpus

Chollet (2019) proposed the Abstraction and Reasoning Corpus (ARC) as a domain for evaluating abstract concept understanding and reasoning abilities in both humans and AI systems. ARC consists of a set of analogy problems, exemplified by those given in Figure 1. In particular, each problem consists of a set of *demonstrations*—initial and transformed grids—and one or more *test input* grids. In Chollet's terminology, the demonstrations coupled with the test inputs form a *task* to be solved. To solve a task, an agent needs

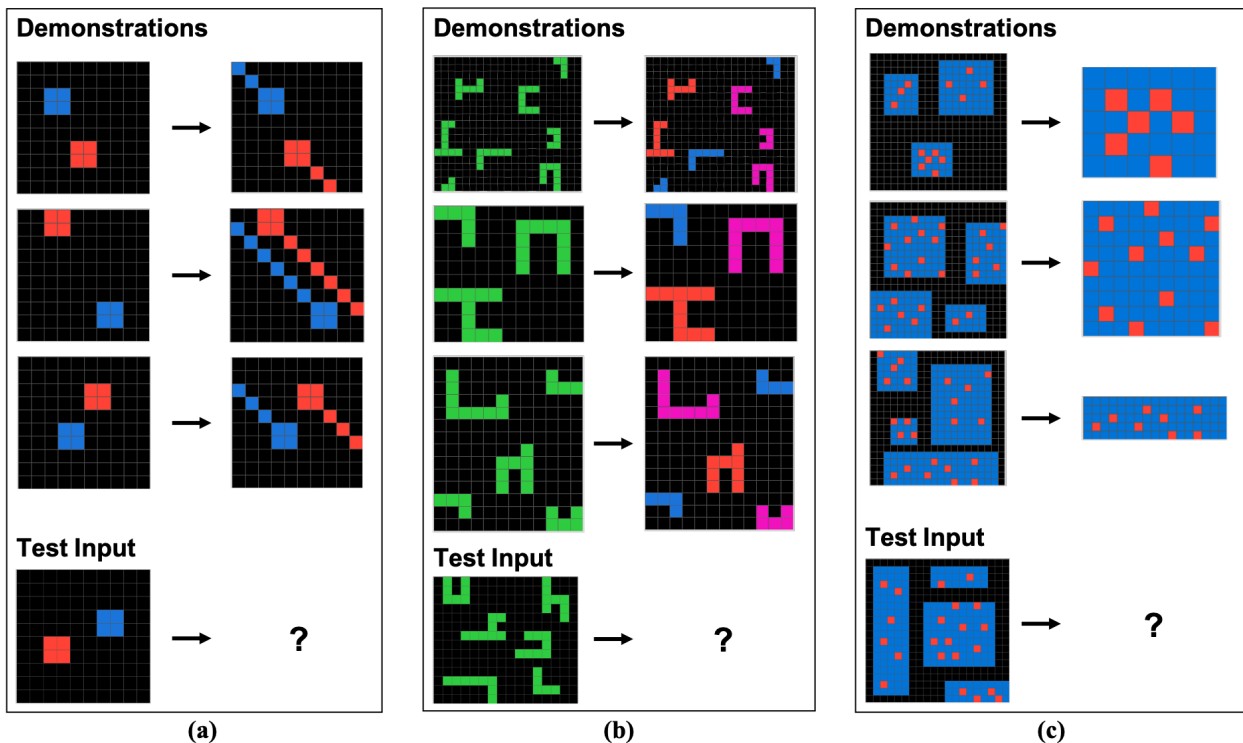

Figure 1: Three sample ARC tasks from Chollet (2023). Each task consists of a set of task demonstrations—transformations between colored grids that follow the same abstract rule—and (here) a single test input. The job of the solver is to generate a new grid that results from applying the abstract rule to the test input. (Best viewed in color.)

to infer the abstract rule governing the demonstrations and apply that rule to each test input to produce a correct output grid.

The ARC domain was inspired by the hypothesis that humans possess innate (or early learned) "core knowledge systems" on which all further learning and knowledge is based. According to Spelke and Kinzler (2007), core knowledge systems include:

(1) **Objectness:** knowledge that the world can be parsed into objects that have certain physical properties, such as traveling in continuous trajectories, being preserved through time, and interacting upon contact;

(2) **Numerosity:** knowledge of small quantities and notions of "smallest," "largest," "greater than," "less than," etc.;

(3) **Basic geometry and topology:** knowledge of lines, simple shapes, symmetries, containment, etc.;

(4) **Agents and goal-directed behavior:** knowledge that some entities are agents who have their own intentions and act to achieve goals.

In creating ARC tasks, Chollet assumed the first three as priors—that is, the only knowledge that should be necessary to solve these tasks. For example, Figure 1(a) requires spatial notions of extending a line diagonally from an object to a boundary; Figure 1(b) requires parsing connected sets of pixels into objects and recognizing shapes across different rotations and symmetries; and Figure 1(c) requires notions of counting and comparisons among quantities.

The tasks in Figure 1 are sampled from the 1,000-task corpus created by Chollet. Eight-hundred tasks were made public online (Chollet, 2023) and as a challenge on the Kaggle platform (Kaggle.com, 2020). The remaining 200 tasks were kept as a "hidden" test set for evaluating AI systems; 100 of these were used to

evaluate submissions to the Kaggle challenge. Each program in the Kaggle challenge was allowed to generate three candidate solutions for each test input in the hidden evaluation set; if one of the three was correct, the test input was considered solved. A task is considered to be solved if all of its test inputs are solved. The first-place program in the Kaggle challenge solved 21% of the 100 hidden tasks; an ensemble of the first- and second-place programs solved about 31%.[1]

ARC remains challenging for AI systems, even for enormous pre-trained language models (see Section 7) for several reasons. ARC tasks involve few-shot learning—inferring an abstract concept from just a few examples. Moreover, the "core knowledge" required is enormously open-ended (e.g., even recognizing an "object" in this domain can require taking context into account), and solving the tasks requires applying core knowledge concepts with a flexibility that is key to human cognition but has not yet been achieved in AI.

One limitation on ARC's usefulness for AI research is that it might be *too* challenging. Many of the tasks in Chollet's corpus are difficult even for humans, and the corpus as a whole might be sufficiently difficult for machines that it does not reveal real progress on machine acquisition of core knowledge. Another limitation is that the current corpus does not systematically test generalization of the concepts underlying individual tasks. For example, if an ARC solver correctly answers the test input in Figure 1(c), one cannot conclude that the solver can generalize the concepts of "counting" and "greater than"—the system might have employed another strategy to solve this specific instance. Only by systematically evaluating a system on many variants of a given concept can we gain evidence that the system grasps that concept in a way that predicts corresponding generalization abilities.

We address these limitations by developing a new benchmark set of tasks in the ARC domain that (1) are designed to rely on straightforward instances of core concepts (and thus be relatively easy for humans), and (2) systematically evaluate the degree to which a task solver has sufficient understanding of a particular concept so as to be able to generalize. Furthermore, we test three programs—the first- and second-place programs from the ARC-Kaggle challenge, as well as OpenAI's GPT-4 pre-trained language model—on our tasks, and compare their performance to humans tested on these same tasks.

## 3 The ConceptARC Benchmark

As a first step in developing new benchmarks for concept understanding in the ARC domain, we created ConceptARC.[2] We studied all publicly available ARC tasks and manually identified 16 concepts used in them (listed in the left column of Table 1). Each of these concepts is central in one or more tasks in Chollet's published ARC "training" and "evaluation" sets, though those sets were not organized around specific concepts, nor does our set of 16 concepts by any means cover all the different concepts used in ARC tasks. For each concept, we created 10 new ARC tasks that are different instantiations of the concept. This set of tasks is termed the *concept group* for a given concept. Each of our tasks has three different test inputs. The purpose of creating multiple test inputs for a given task was to control for possible "shortcuts" that might enable a solver to solve a particular test input without correctly grasping the underlying rule, whereas the purpose of creating multiple tasks within a concept group was to ensure that a solver is able to generalize across different instantiations of the concept.

As an example, Figure 2 shows three tasks from ConceptARC that are variations on the concept *Sameness*. Figure 2(a) focuses on sameness between shapes (in each transformation, only objects with the same shape are retained); in Figure 2(b) lines with the same orientation are retained, and in Figure 2(c) each grid is divided (by a gray line) into two subgrids; if the two subgrids are identical, both are copied, and if not, only the lefthand subgrid is copied. These sample tasks illustrate the range of variation among test inputs for a given task and among tasks in a given concept group. This range is meant to be sufficiently broad that an agent that correctly solves most or all of the tasks in a group is likely to possess a rich understanding of the concept. (Examples of problems from each concept group are given in Appendix A.)

---

[1] Nearly all of the 800 published tasks had only one test input; the number of test inputs per task in the hidden test set was not revealed.

[2] All ConceptARC tasks can be downloaded from `https://github.com/victorvikram/ConceptARC`.

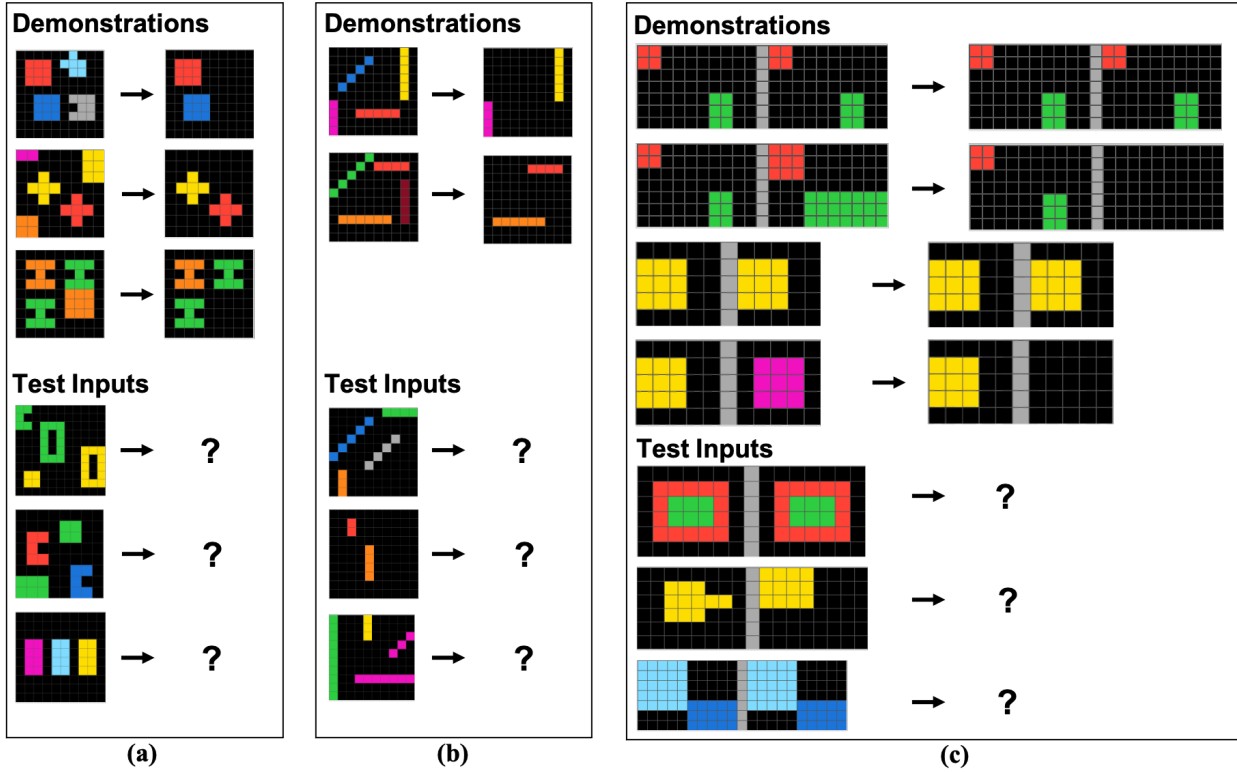

Figure 2: Three sample tasks from ConceptARC, each of which has a set of demonstrations and three test inputs. Each task is a variation on the concept *Sameness*. (Best viewed in color.)

We constructed the tasks in the ConceptARC benchmark manually.[3] We do not believe that interesting, diverse task variations on a particular concept could be constructed automatically, unless we were able to create an automated system that understands the concept in a general way (and the challenge of developing such a system is what inspired the benchmark in the first place). Given that the goal of the ARC benchmark is to evaluate humanlike concept abstraction, we (following Chollet (2019)) constructed the tasks using our own intuitions about human core concepts.

It is important to note that "concept" does not have a rigorous definition in cognitive science. Here we use the term informally to refer to a facet of core knowledge that plays important roles in the original ARC tasks, and that play more obviously central roles in the ConceptARC tasks in a given concept group.

## 4    Human and Machine Performance on ConceptARC

In this section we present results from our studies of human and machine performance on ConceptARC. We recruited human participants using the Amazon Mechanical Turk and Prolific platforms and tested them on tasks in our corpus, using a visual interface. We also obtained code for the first- and second-place ARC-Kaggle winning programs and ran them on the same tasks, in the same text-based format used in the ARC-Kaggle competition. Finally we used OpenAI's API to test GPT-4 on these tasks, using a text prompt similar to the format given to the ARC-Kaggle programs. Details of each study are given in the next sections.

Recall that each of our 16 concept groups contains 10 tasks, each of which includes three unique test inputs, for a total of 30 test inputs per concept. Both humans and machine solvers are allowed three guesses for

---

[3]It should be noted that in our tasks, following human conventions, the color black plays a special role, signifying background or "unfilled" grid squares.

each test input, and a solver (human or machine) is considered correct on a test input if one of the three guesses is correct.

Table 1 gives, for each concept, the accuracies over the 30 test inputs in the concept group. These results provide an assessment of how well solvers can generalize over the range of different tasks associated with each concept. The human accuracy reported for each concept is the average accuracy over the 30 test inputs in that concept-group, where the accuracy on a given test input is the fraction of participants who correctly solved that test input. The accuracies reported for each Kaggle-ARC program and for GPT-4 are simply the fraction of test inputs in each concept group that were correctly solved by the program. We discuss these results in detail in Section 8.[4]

Table 1: Accuracies of humans, the two top-scoring ARC-Kaggle programs, and GPT-4 (with temperatures 0 and 0.5) on test inputs in each concept group in ConceptARC.

| Concept | Humans | ARC-Kaggle First Place | ARC-Kaggle Second Place | GPT-4 Temp 0 | GPT-4 Temp 0.5 |
|---|---|---|---|---|---|
| Above and Below | 0.90 | 0.70 | 0.33 | 0.23 | 0.37 |
| Center | 0.94 | 0.50 | 0.20 | 0.33 | 0.33 |
| Clean Up | 0.97 | 0.50 | 0.20 | 0.20 | 0.27 |
| Complete Shape | 0.85 | 0.47 | 0.30 | 0.23 | 0.23 |
| Copy | 0.94 | 0.23 | 0.27 | 0.23 | 0.27 |
| Count | 0.88 | 0.60 | 0.40 | 0.13 | 0.17 |
| Extend To Boundary | 0.93 | 0.77 | 0.47 | 0.07 | 0.1 |
| Extract Objects | 0.86 | 0.43 | 0.43 | 0.03 | 0.07 |
| Filled and Not Filled | 0.96 | 0.73 | 0.43 | 0.17 | 0.27 |
| Horizontal and Vertical | 0.91 | 0.43 | 0.10 | 0.27 | 0.33 |
| Inside and Outside | 0.91 | 0.57 | 0.10 | 0.10 | 0.16 |
| Move To Boundary | 0.91 | 0.37 | 0.30 | 0.20 | 0.20 |
| Order | 0.83 | 0.27 | 0.23 | 0.27 | 0.27 |
| Same and Different | 0.88 | 0.53 | 0.17 | 0.17 | 0.27 |
| Top and Bottom 2D | 0.95 | 0.60 | 0.57 | 0.23 | 0.37 |
| Top and Bottom 3D | 0.93 | 0.50 | 0.03 | 0.20 | 0.27 |

# 5 Details of Human Studies

To evaluate human accuracy in our tasks, we ran an online study, recruiting participants from the Amazon Mechanical Turk and Prolific platforms. This section provides additional details on our procedure.

## 5.1 Procedure

Participants were presented with a visual interface for solving ARC tasks, adapted from the original ARC viewer (Chollet, 2023), and programmed for online data collection using the psiTurk framework.[5] Each participant was presented with a random selection of tasks (17 for most participants, although see Appendix C for a discussion of a few exceptions). Each task had three test inputs, but these were randomly split among participants, with each participant seeing only one test input of a given task. Similar to the ARC-Kaggle programs, participants were given three attempts to solve each test input. If a participant managed to solve the test input correctly, they were asked to verbally describe their solution before moving on to the next task. We will report on the analysis of this natural language data in future work.

Note that among the 17 tasks given to a participant, the first two were extremely simple training tasks, for which the participants were allowed unlimited attempts. These training tasks were included to give the participants time to familiarize themselves with the study interface. Additionally, among the remaining tasks, three were "minimal," that is, the simplest concept instantiations we could create. These minimal tasks served as "attention checks," helping to identify individuals who did not try to solve the tasks or follow instructions (see Section 5.2). We give examples of minimal tasks in Appendix B. Because they were used to

---

[4]Results for human participants and machines on all 480 test inputs can be downloaded from `https://github.com/victorvikram/ConceptARC`.

[5]`https://psiturk.org/`

determine which participants to exclude, we did not include performance on the minimal tasks in the results given in Section 4.[6]

## 5.2 Exclusion criteria

We used two criteria to exclude participants. A participant was excluded from further analysis if 1) they failed at solving two or more minimal tasks; or 2) they provided empty or nonsensical explanations for their solutions (such as "Nice," "Solve task is good," and so on). Failing the first criterion suggests that the person was not paying attention to the task, while failing the second shows lack of ability or motivation to follow the task instructions. Since it was always faster for a participant to pretend to fail any given problem rather than to try to solve it, excluding unmotivated, inattentive participants was crucial to avoid skewing the results.

In total, 55 out of 482 initial participants were excluded based on inadequate explanations (all from Amazon Mechanical Turk). An additional 12 participants were excluded based on failing to solve two or more minimal tasks (8 from Amazon Mechanical Turk, 4 from Prolific).

## 5.3 Participants

The final sample comprised 415 participants—204 from Amazon Mechanical Turk and 211 through Prolific. To ensure linguistic fluency in English for the purpose of collecting natural language descriptions, only U.S.-based Amazon Mechanical Turk workers were invited to participate in the study, and only U.S. or U.K. participants were recruited from Prolific.[7]

Since test inputs were randomly assigned to different participants, and since the psiTurk platform does not naturally have a mechanism to monitor how many participant answers were collected for each test input, there is some variation in the amount of data collected for different test inputs. Overall, each test input was given to at least 8 participants (with one exception, which was given to only 7 participants). The detailed results available at `https://github.com/victorvikram/ConceptARC` provide the number of participants tested on each test input in the corpus.

## 6 Details of Testing Winning Programs From the ARC-Kaggle Challenge

As we described in Section 2, in 2020 the Kaggle platform hosted a three-month competition on ARC tasks (Kaggle.com, 2020). Competing programs were scored on 100 hidden tasks. Programs were allowed to make three predictions for each test input, and if one of the predictions was correct, the test input was considered to be solved. Using this metric, the first and second place programs attained accuracies of 21% and 19% respectively. An ensemble of the two winning programs attained an accuracy of about 31%, and as of this writing, this is the state-of-the-art accuracy on this hidden evaluation set.[8] To our knowledge, there have been no published large-scale experiments to date evaluating humans on tasks in the ARC corpus (though, as we describe in Section 8, small-scale studies were performed by Acquaviva et al. (2022) and Johnson et al. (2021)).

We obtained the source code for the first- and second-place ARC-Kaggle winners on GitHub.[9], and tested each of them on all of the ConceptARC tasks.

The first and second place programs in the ARC-Kaggle challenge both work by performing a heuristic search over a fixed, manually defined set of grid operations to generate a pipeline of these operations that, when applied to inputs from the task demonstrations, correctly generates the corresponding outputs.

---

[6]All of the minimal tasks are included in the corpus provided at `https://github.com/victorvikram/ConceptARC`.

[7]We cannot exclude the possibility that a person from another country might register a U.S.-based account on Prolific or Amazon Mechanical Turk. However, we have manually checked the verbal answers provided by the study participants. All participants in the final sample demonstrated high fluency in English, which at least should ensure that they fully understood the study instructions.

[8]F. Chollet, Personal Communication, April 7, 2023.

[9]`https://github.com/top-quarks/ARC-solution` (first-place ARC-Kaggle winner); `https://github.com/alejandrodemiquel/ARC_Kaggle` (second-place ARC-Kaggle winner).

In particular, the first-place program (Wind, 2020a) constructed its solutions using a manually created set of 142 grid operations, such as an operation that splits a given grid into multiple grids consisting of "background color" and "objects" consisting of color-connected pixels, operations that perform rotations, reflections, and other variations on a given grid, and an operation that extracts the "object" with the most non-black pixels. The second-place program (de Miquel Bleier, 2020) constructed its programs from a set of 50 manually created grid operations, and used a genetic algorithm to search for successful pipelines of operations.

Both programs were able to increase their success by augmenting the given task demonstrations, for example, by flipping the demonstration input and output grids along the diagonal, by remapping colors, and other heuristic transformations.

Given the open-ended nature of ARC, we doubt that similar heuristic search methods, even over a much larger number of grid operations, will achieve anything like human performance on ARC tasks. Even the authors of the winning programs seem to agree that a wholly different kind of method is needed. The author of the first-place program wrote, "Unfortunately, I don't feel like my solution itself brings us closer to AGI" (Wind, 2020b) and one of the authors of the second-place program noted that "No team out of the 914 [ARC-Kaggle competition] participants found a satisfying, AI-focused solution for this problem" (de Miquel Bleier, 2020).

# 7 Details of Testing GPT-4

GPT-4 (OpenAI, 2023) is a large-scale multimodal AI system created by OpenAI. Webb et al. (2022) showed that the publicly available language-only version of GPT-4 (as well as its predecessor GPT-3) was able to match or exceed human performance on several idealized analogy tasks, in a zero-shot manner (i.e., without any fine-tuning on these tasks). To test the generality of these findings, we assess GPT-4's zero-shot performance on the tasks in ConceptARC, which have some resemblance to the tasks used by Webb et al.

To test this language-only version of GPT-4 on the tasks in ConceptARC, we used the API provided by OpenAI.[10] GPT-4 API prompts have "system" and "user" components, with the "system" component intended to provide general instructions, priming the model towards certain behaviors, and the "user" component for dialogue inputs. We used the prompt structure (similar to the one used by Webb et al. (2022)) illustrated in Figure 3.

Within each row of a grid, the colors of each pixel were numerically coded as in the original ARC data files given by Chollet (2023) (these were the inputs to the ARC-Kaggle competitors) and space-separated. For example, [2 1 0 1] would encode a row with four pixels: red, blue, black, and blue again.

We tested GPT-4 on all the tasks in our corpus, first with temperature 0 (following Webb et al. (2022)) and then with temperature 0.5. In every case, GPT-4's response was in the correct format for an output—that is, the errors that the model made were "true" mistakes, rather than improperly formatted correct answers. In the case where temperature was set to 0, the outputs are deterministic, so only one output was considered. In the case where temperature was set to 0.5, we repeated each task prompt three times, and if at least one of outputs was correct, we considered the task to be solved correctly.[11]

# 8 Discussion

## 8.1 Human and Machine Performance

Table 1 shows that the human participants achieved substantially higher accuracy than the machine solvers on tasks in our ConceptARC benchmark (for a more detailed report with uncertainty estimates, see Appendix D). Recall that the average accuracy across test inputs in a concept group measures how well solvers

---

[10]`https://openai.com/product`. In our experiments the model name was set to "gpt-4", the temperature was set to 0 or 0.5, and other parameters were left at their default values.

[11]Following Webb et al. (2022), we also tested GPT-4 using the same prompt format as in Figure 3 but with '\n' inserted after each grid row, to indicate new lines. The results were not substantially different from testing GPT-4 without inserting '\n'.

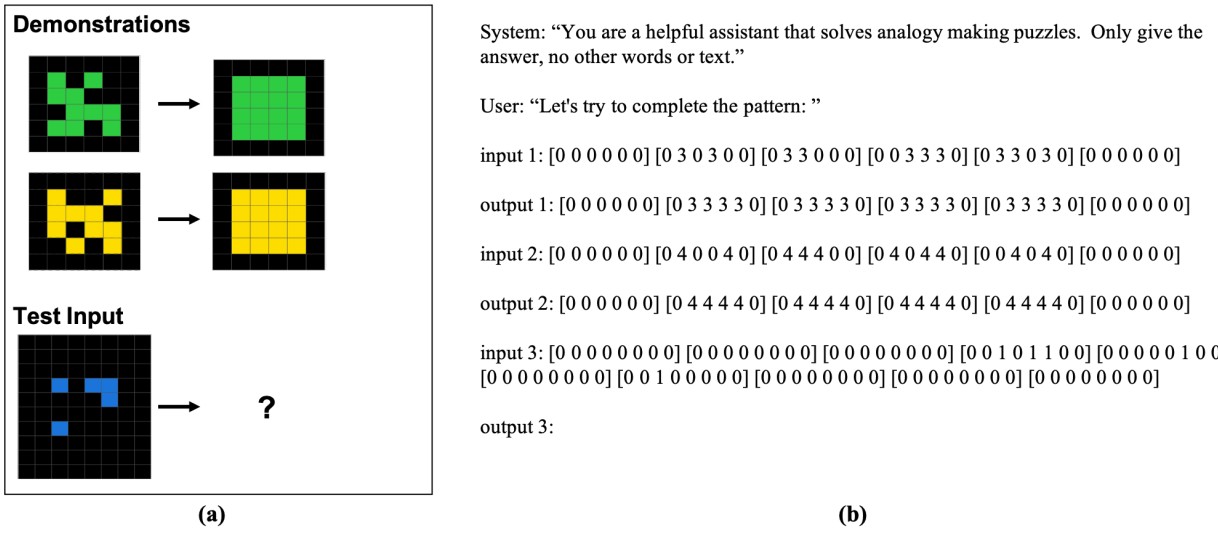

**Demonstrations**

**Test Input**

**(a)**

System: "You are a helpful assistant that solves analogy making puzzles. Only give the answer, no other words or text."

User: "Let's try to complete the pattern: "

input 1: [0 0 0 0 0 0] [0 3 0 3 0 0] [0 3 3 0 0 0] [0 0 3 3 3 0] [0 3 3 0 3 0] [0 0 0 0 0 0]

output 1: [0 0 0 0 0 0] [0 3 3 3 3 0] [0 3 3 3 3 0] [0 3 3 3 3 0] [0 3 3 3 3 0] [0 0 0 0 0 0]

input 2: [0 0 0 0 0 0] [0 4 0 0 4 0] [0 4 4 4 0 0] [0 4 0 4 4 0] [0 0 4 0 4 0] [0 0 0 0 0 0]

output 2: [0 0 0 0 0 0] [0 4 4 4 4 0] [0 4 4 4 4 0] [0 4 4 4 4 0] [0 4 4 4 4 0] [0 0 0 0 0 0]

input 3: [0 0 0 0 0 0 0] [0 0 0 0 0 0 0] [0 0 0 0 0 0 0] [0 0 1 0 1 1 0 0] [0 0 0 0 0 1 0 0] [0 0 0 0 0 0 0 0] [0 0 1 0 0 0 0 0] [0 0 0 0 0 0 0 0] [0 0 0 0 0 0 0 0] [0 0 0 0 0 0 0 0]

output 3:

**(b)**

Figure 3: (a) ConceptARC task. (b) Corresponding prompt given to GPT-4. (Best viewed in color.)

can generalize over different tasks representing a given concept. The average difference in per-concept accuracy between humans and the first-place ARC-Kaggle program was 40 percentage points. The human participants exhibited over 90% average accuracy on 11 of the 16 concepts, and over 80% accuracy on each of the remaining 5 concepts. In contrast, the first-place program never scored above 80% accuracy on any concept, and for 11 out of 16 concepts, its accuracy was below 60%. The second-place program's accuracy never reached 60% and was below 50% on 15 out of 16 concepts. On each concept, GPT-4 performed better with temperature set to 0.5 (and given three guesses) than it did with temperature set to 0. But even in the former case GPT-4's performance was relatively weak: its accuracy was below 30% on 12 out of 16 concepts and its maximum accuracy was only 37%.

GPT-4's weak performance here contrasts with its much better performance on other idealized domains for analogy-making found by Webb et al. (2022). We speculate that there are two potential reasons for this discrepancy. First, to successfully solve ConceptARC problems, a system must identify the relevant higher-level features (e.g., objects, shapes, number, orientation) in each grid and determine how they were transformed into a new grid. In the grid-based tasks discussed in Webb et al., which were inspired by Raven's Progressive Matrices, the relevant features were pre-extracted and given as inputs. This likely makes the grid-based tasks created by Webb et al. considerably less challenging than those in ConceptARC. Second, we speculate that the patterns in Raven-based tasks tested by Webb et al. might be easier to recognize for large language models: each grid row had a constant number of features (which might aid in finding analogies using attention mechanisms) and the concepts in these tasks were generally of a more sequential rather than spatial nature. Testing these speculations is a topic for future work.

The generally high accuracies of humans on each concept indicates successful generalization over the different variations in each given concept group. In contrast, the much lower accuracies of programs we tested indicates a lack of ability to generalize over the variations in a concept group, and thus a failure to develop the abstractions that ARC is meant to test.

While the first- and second-place ARC-Kaggle programs did not reach human-level accuracy, it is interesting to note that these programs have significantly higher accuracy on the tasks in ConceptARC than they did on the original tasks in the ARC-Kaggle competition, where their respective accuracies were 21% and 19%. This is likely due to our intentional design of the tasks in ConceptARC to be easier than those in the original ARC set. Providing an easier benchmark also gives more insight into differences between the two programs: while their scores on the original set were quite close, in our results the first-place winner has substantially

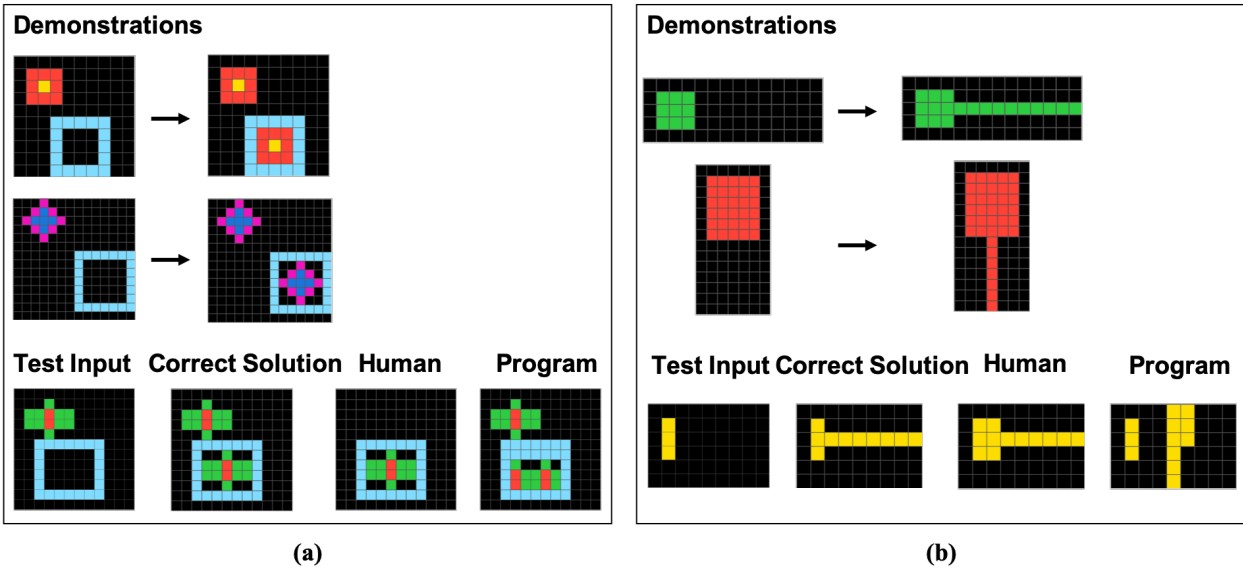

Figure 4: Two examples illustrating human "near-miss" errors, compared with errors made by the first-place ARC-Kaggle program on the same test input. **(a)** A task in the *Copy* concept group. The human correctly copied the green and red object into the blue rectangle, but incorrectly deleted the original object. The first-place program ("Program") did not seem to grasp the notion of copying an object. **(b)** A task in the *Extend To Boundary* concept group. The human correctly extended a line to the boundary, but modified the original object to make it a solid rectangle rather than a single line. The first-place program did not seem to grasp the notion of extending a line from a given object to a boundary. (Best viewed in color.)

higher accuracy than the second-place winner—the average difference in per-concept accuracy between the two programs is about 23 percentage points.

We ran a simple $\chi^2$ test to see whether there were statistically significant differences in accuracy across concepts. For humans, as well as for Kaggle First- and Second- place algorithms, the data strongly suggest such a difference (p < 0.001 in each of the three hypothesis tests). At the same time, GPT-4 results did not provide sufficient evidence that there were true differences in its mastery of those concepts ($p = 0.27$ and $p = 0.23$ for 0 and 0.5 temperature regimes respectively). In other words, the observed variation in GPT-4's performance across concepts might have occurred by chance. At the same time, GPT-4's generally low accuracy across all concepts, together with the relatively low-data regime, limits the statistical power of this test, hence this negative result should be interpreted with caution.

### 8.2 Comparing Human and Machine Errors

It is enlightening to compare the kinds of errors made by humans and those made by programs on these tasks. In analyzing a sample of errors made by the human participants in our study, we found that many errors included obvious careless mistakes (e.g., off-by-one errors in the size of the output grid), wrong answers due to "giving up" (simply copying the input grid or creating a blank grid), and "near misses," in which it is obvious that the person grasped the underlying concept but made an error in applying it. In Figure 4 we show two examples of these types of near-misses by humans, and corresponding incorrect answers to the same task by the first-place ARC-Kaggle program.

As illustrated in Figure 4, the errors made by the winning ARC-Kaggle programs and by GPT-4 are harder to categorize. As we described in Section 6 above, the ARC-Kaggle winning programs were not designed to capture abstract *concepts*, but instead heuristically construct pipelines of pre-designed grid transformations, so it is not surprising that their errors were typically less interpretable than those of humans. GPT-4 of course

was not designed for this domain at all, although Webb et al. (2022) demonstrated the pattern-recognition abilities of large language models in other analogy domains.

### 8.3 Limitations of Our Studies

There are several limitations of the studies we report here in using the ConceptARC corpus to assess conceptual abstraction abilities in humans and machines.

Because the tasks in ConceptARC were created manually, the corpus is relatively small: 16 concept groups, with 10 tasks per concept-group and three test inputs per task, for a total of 480 test inputs. We plan to substantially extend this corpus in the future, adding additional concept groups, tasks, and test inputs in order to more thoroughly explore abstraction and generalization abilities in the ARC domain.

Our human studies, using Amazon Mechanical Turk and Prolific, included 415 participants, each being tested on approximately 17 test inputs (from different tasks) in an approximately 45-minute session. As we described in Section 5, this yielded typically 8 to 14 people solving each test input. Our results, showing high human accuracy on these tasks, are based on these relatively small sets of people, whose numbers were limited by the funds we had available for these studies. In future work we will extend these studies to determine if they generalize over larger populations of human solvers.

One additional limitation—for both ConceptARC and the original ARC dataset—is that there might be more than one reasonable solution for a given test input. Like Chollet in the original ARC set, we tried to design tasks that have only one clear solution for each test input. However, there is always the possibility of other solutions that people would find reasonable, but that would be counted as incorrect in our study. By closely examining incorrect solutions submitted by humans in our studies, we found six (out of 480 total) test inputs that we considered to be ambiguous in this way. There might be additional such test inputs that we did not identify. Indeed, a small number of ambiguous test inputs are likely inevitable in any corpus. However, the effects of such ambiguity on our results are mitigated, for both humans and machines, by (1) allowing three solutions to be submitted for each test input; (2) having multiple test inputs per task; and (3) having multiple tasks per concept group. These factors help make the aggregate results robust in spite of an inevitable small subset of ambiguous test inputs.

There are also a small number of tasks that allow for "shortcut solutions": for example, tasks in which the correct solution to a test input is simply to copy it, which can be a default strategy for the ARC-Kaggle winners and an easy pattern for GPT-4 to recognize. However, our purpose in creating numerous tasks that are variations on a particular concept is to make it highly unlikely that any program could use shortcuts to solve most or all of the tasks in a given concept group.

## 9 Related Work

In a similar spirit to our work on the ConceptARC benchmark, Kim et al. (2022) created the "Mini-ARC" dataset, in which grids are fixed at $5 \times 5$ in order to simplify the domain, and 150 tasks (each containing one test input) are organized around six broad categories (movement, color, object, number, geometry, and "common sense"). This set can complement our ConceptARC benchmark, which allows any grid dimensions and systematically explores 16 more specific spatial and semantic concepts.

Johnson et al. (2021) carried out a study of humans solving ARC tasks. They chose 40 tasks from the public ARC dataset and tested each of 95 participants on a random subset of 10 out of the 40 tasks. On average the participants' per-task accuracy was about 84%, though with substantial variance. Johnson et al. also recorded the average time to complete each task, as well as participants' action sequences while generating responses, and analyzed the errors people made. Similar to the results of our study, the authors found that human errors generally were near-misses, whereas the errors made by the first-place ARC-Kaggle program indicated that it did not grasp the underlying abstract rule. The human study we report in this paper can be seen as a follow-up to Johnson et al.'s study, but with a larger set of (newly created) ARC problems that are variations on specific concepts (rather than a randomly chosen subset of the original ARC tasks) and with a considerably larger population of participants.

In developing AI systems to solve ARC problems, the predominant approach is automated program synthesis—that is, automatically generating a series of operations on grids or or other representations that yields a solution. The primitive operations are typically created manually, and heuristic search is used to find a combination of operations that solves a given task. For the two winning ARC-Kaggle programs, the primitive operations were sets of hand-designed grid transformations, and the synthesized programs were pipelines of transformations resulting from heuristic search methods.

Since the end of the Kaggle competition, several new program-synthesis approaches to ARC have been explored. For example, Banburski et al. (2020) used a program-synthesis algorithm called "DreamCoder" (Ellis et al., 2020) that, given a domain-specific primitive operation, can generate a more abstract operation that can be added to the set of available primitives. Banburski et al. manually defined a small set of grid-transformation operations, and used these as the basis for training an agent based on DreamCoder to generate programs that could solve a small set of ARC tasks that focused on symmetry transformations. In follow-up work, Alford et al. (2022) explored a similar method using a neural-network-guided program-synthesis approach.

In contrast to using grid-transformation primitives, Xu et al. (2022) proposed a "object-centric" approach to solving ARC tasks. In their system, grids are mapped to graph representations, and the system searches for pipelines of operations on these graphs rather than on the grids themselves. The nodes in a graph correspond to objects in a grid and links between nodes correspond to relationships between objects. Which sets of pixels are grouped as an "object" in a graph is decided heuristically, as is what relationships are included in the graph. In their experiments, Xu et al. focused on a set of 160 "object-centric" tasks from the public ARC dataset, and showed that their system was able to solve about a third of them.

In an interesting cognitive-science-based study, Acquaviva et al. (2022) posited that the advantage of humans on ARC tasks may be due to their ability to generate descriptions of abstract concepts in natural language. The authors carried out a study, like ours, in which they asked human participants to both solve ARC problems and generate natural-language instructions that would enable another human to produce the correct output, given only the test input (i.e., not including the demonstrations). The authors then tested these instructions on other human participants and found that the instructions were sufficient for solving the task about 88% of the time. The authors released the LARC (Language-Complete ARC) dataset, which couples 354 original ARC tasks with human-generated language instructions. They used this dataset to train and evaluate selected program-synthesis methods, to see if these systems could utilize language the way humans do. The results were quite poor—the best system was able to solve only about 12% of the tasks it was tested on.

Any of these program-synthesis systems might be improved by adopting more expressive domain-specific languages and by improving their program-search methods. While these and other approaches to ARC-like tasks (Ainooson et al., 2023; Assouel et al., 2022; Ferré, 2021; Fischer et al., 2020) have produced some promising results, as of this writing the first-place ARC-Kaggle program remains the most successful single approach (though as we described above, an ensemble of the top two winning programs attained higher accuracy). As yet, there is no AI system that is close to reaching human accuracy and generalization abilities on ARC tasks. The ARC domain remains a wide-open challenge for AI.

## 10 Conclusions and Future Work

In this paper we have described ConceptARC, a new benchmark set of tasks in the ARC domain. The tasks in ConceptARC are designed to systematically test conceptual understanding and generalization while remaining relatively easy for humans. Our purpose in designing a benchmark with these attributes is threefold: first, to promote the development of AI systems that grasp generalizable core concepts and are able to use them in new situations; second, to fairly evaluate systems that are claimed to have such abilities; and third, to provide an evaluation set that is not overly difficult, and that would thus mask real progress in developing such systems.

In addition to describing and publishing the ConceptARC benchmark, we have reported results of testing humans and machine solvers on these tasks. Our results show that humans substantially outperform state-

of-the-art programs on all the concepts in our benchmark; moreover, when humans make errors, they often still exhibit a grasp of the underlying concept, unlike the programs. We also showed that our benchmark is able to reveal differences in performance among machine solvers that were masked by the difficulty of the original ARC dataset. In addition, we showed that GPT-4's performance, while impressive given that in many cases it exceeded the second-place ARC Kaggle program even though it was not designed for or trained on these tasks, is dramatically below that of humans, which contrasts with the results of Webb et al. (2022) in testing GPT-4 on other idealized analogy domains.

As we described in Section 5, in addition to asking participants to solve tasks, we also asked them to write natural language instructions for solving a given test input. In the near future we will perform a new study with human participants to test the viability of these instructions, by giving a test input (without accompanying demonstrations) along with the corresponding instructions, to see if people can arrive at the correct solution by following these instructions. Following Acquaviva et al. (2022), we will take the viable instructions and use them as part of a training set for a new machine ARC solver, to see if augmenting training with language inputs will improve performance.

In the future we also plan to extend the ConceptARC benchmark to encompass additional tasks and concept groups, and to further evaluate humans and machine solvers on these new tasks, as well as to more thoroughly analyze the different types of errors made by humans and machines. In particular, in addition to the tasks that we make publicly available, we will create a "hidden" evaluation set that can be used in future ARC competitions.

When solving a task in the ARC domain, humans bring to bear not only their core knowledge about the world but also a highly evolved visual system that is not present in any of the proposed machine solvers or in GPT-4. While the grids in an ARC task are visually simple, it may be that incorporating routines inspired by the visual system (Ullman, 1987) into program-synthesis approaches could be a way to make progress on these tasks. We plan to explore this hypothesis in future work. We also plan to test the multimodal version of GPT-4 on ARC tasks, once it is made publicly available.

### Acknowledgments

This material is based in part upon work supported by the National Science Foundation under Grant No. 2139983. Any opinions, findings, and conclusions or recommendations expressed in this material are those of the authors and do not necessarily reflect the views of the National Science Foundation. This work has also been supported by the Templeton World Charity Foundation, Inc. (funder DOI 501100011730) under the grant `https://doi.org/10.54224/20650`.

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

## Appendix A  Examples of Tasks From Each Concept Group

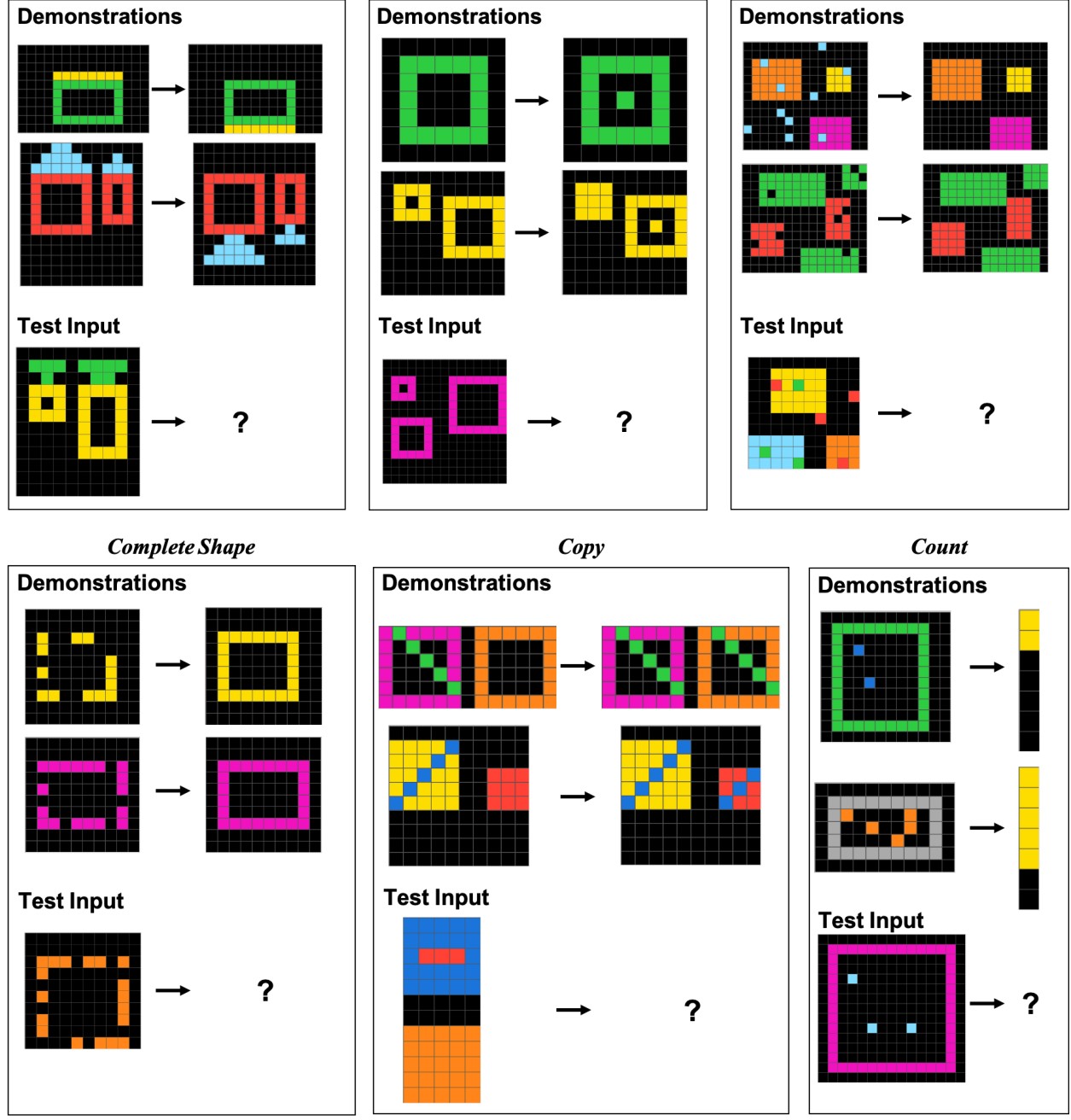

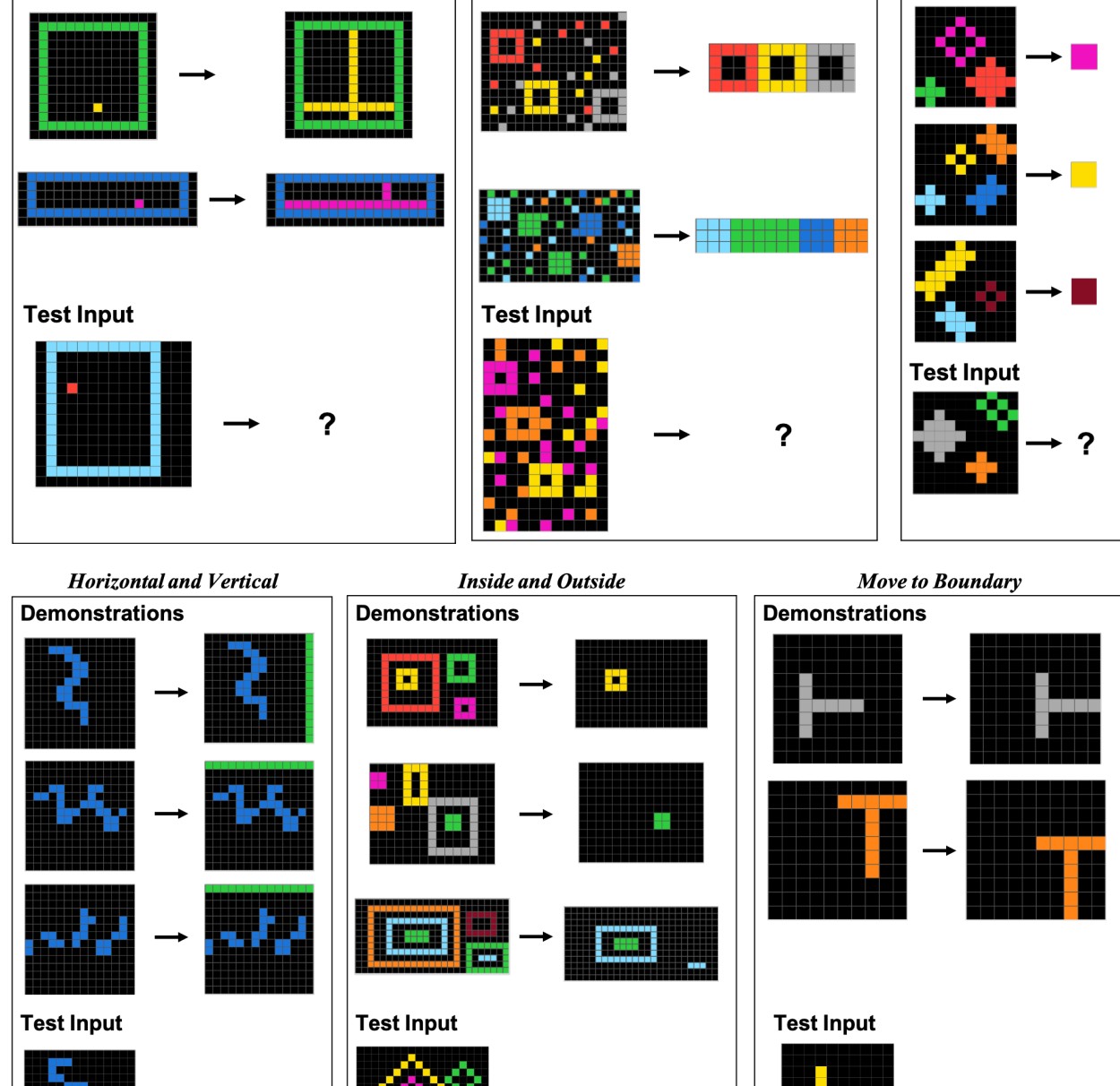

*Order*

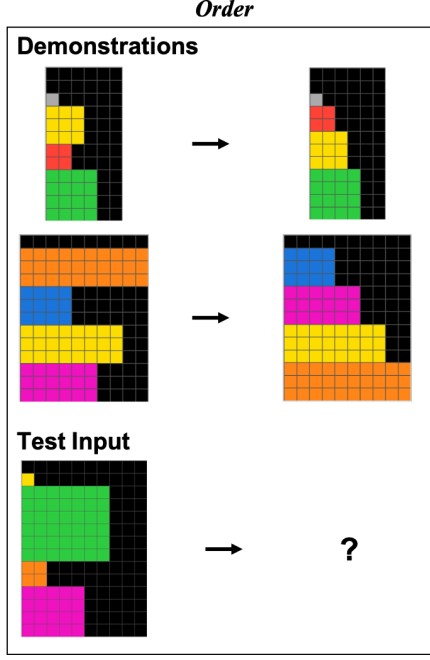

*Same and Different*

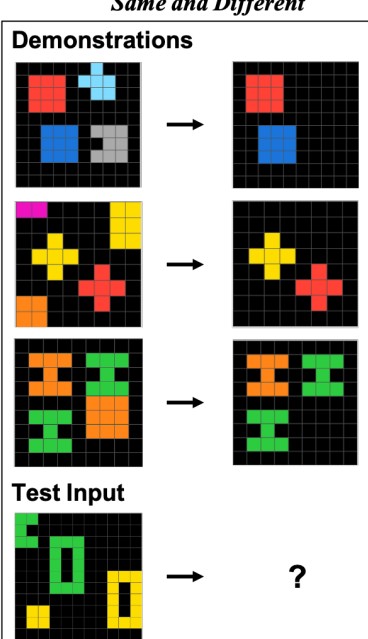

*Top and Bottom 2D*

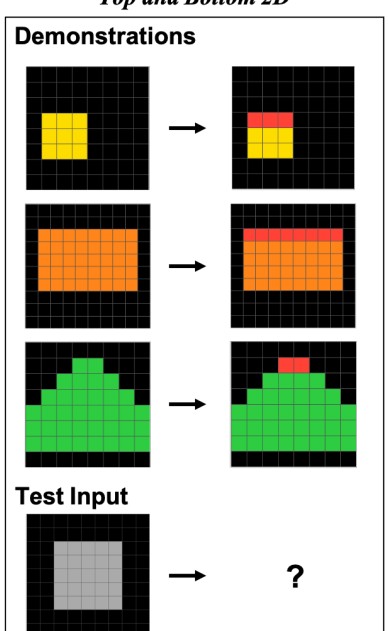

*Top and Bottom 3D*

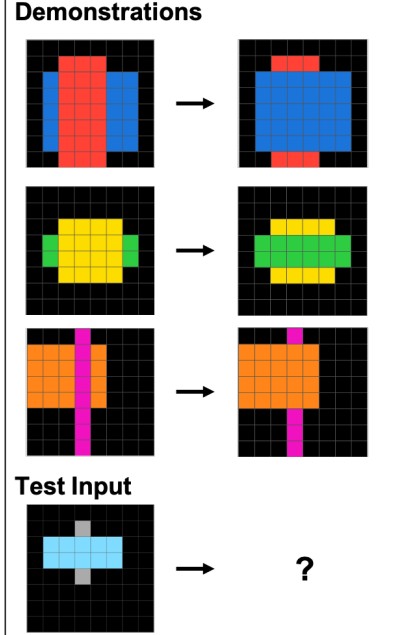

## Appendix B    Examples of Minimal Tasks

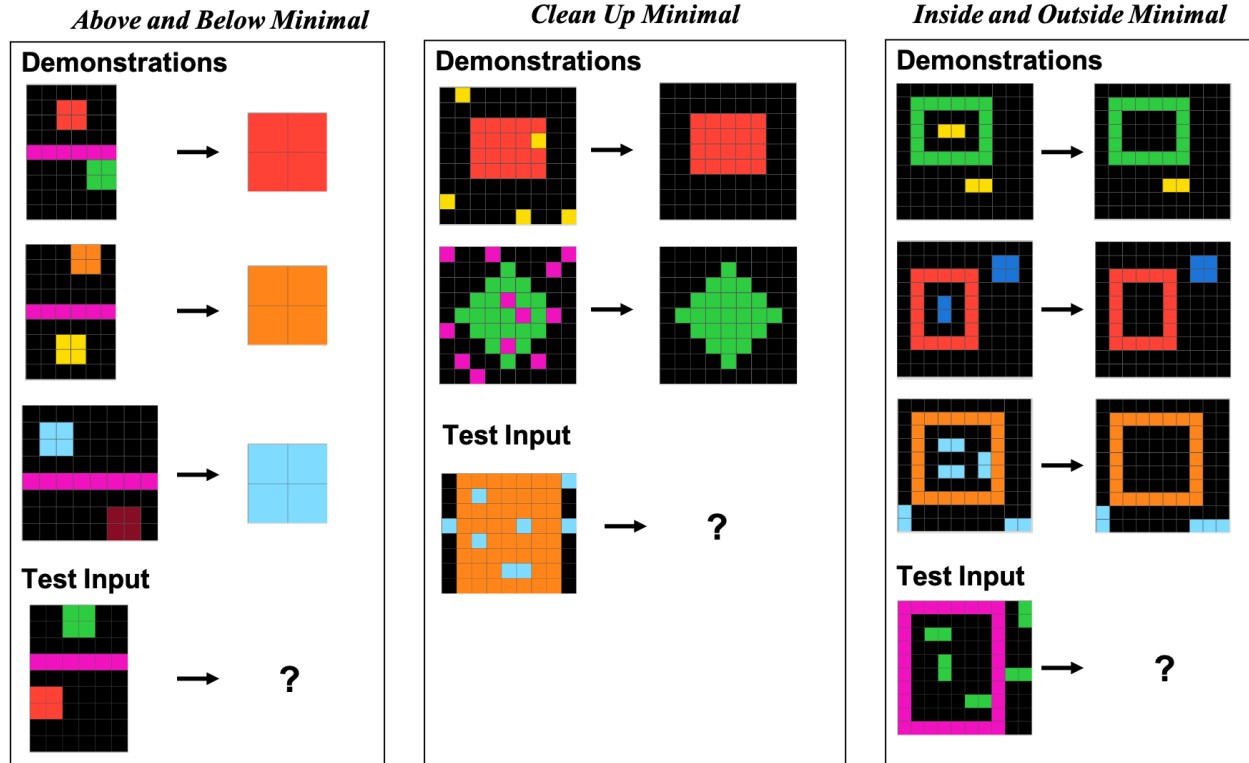

## Appendix C    Participant Recruitment Details in Our Human Study

In order to establish the data collection regime that yields the highest quality data, we introduced minor recruitment and procedure adjustments after the start of data collection.

In the first batch of participants collected via Amazon Mechanical Turk, each received 11 problems (this batch also only had two "minimal Problems," as opposed to three such problems for everyone else). However, preliminary data examination showed that some participants did not fully follow the study instructions and had to be excluded (see Section 5.2). In response, we made the screening criteria more strict (requiring a Master Worker qualification, 99% of HITs approved with at least 2000 HIT history, as opposed to 95% approval requirement in the first batch). Participants in all but the first batch were paid $10 upon completing the experiment. Participants in the first batch were paid $5. In all batches, the median pay-per-hour exceeded the U.S. minimal wage.

Additionally, since participant quality was very "bimodal" (i.e. each participant either diligently followed instructions on all tasks, or ignored instructions on all tasks and were thus excluded), we increased the number of tasks per participant, so that non-excluded participants provided us with more data.

Lastly, after the first large batch of participants, we transitioned the study to another crowdsourcing platform: Prolific.org. This was done both due to technical reasons and to diversify the data source.

# Appendix D    Accuracies With Binomial Confidence Intervals

Table 2: Accuracies of humans, the two top-scoring ARC-Kaggle programs, and GPT-4 (with temperatures 0 and 0.5) on test inputs in each concept group in ConceptARC (same as Table 1), with binomial confidence intervals added.

| Concept | Humans | ARC-Kaggle First Place | ARC-Kaggle Second Place | GPT-4 Temp 0 | GPT-4 Temp 0.5 |
|---|---|---|---|---|---|
| Above and Below | 0.90 (0.86, 0.93) | 0.70 (0.52, 0.83) | 0.33 (0.19, 0.52) | 0.23 (0.12, 0.41) | 0.37 (0.22, 0.54) |
| Center | 0.94 (0.90, 0.96) | 0.50 (0.33, 0.67) | 0.20 (0.1, 0.37) | 0.33 (0.19, 0.51) | 0.33 (0.19, 0.51) |
| Clean Up | 0.97 (0.94, 0.99) | 0.50 (0.33, 0.67) | 0.20 (0.1, 0.37) | 0.20 (0.1, 0.37) | 0.27 (0.14, 0.44) |
| Complete Shape | 0.85 (0.8, 0.89) | 0.47 (0.3, 0.64) | 0.30 (0.17, 0.48) | 0.23 (0.12, 0.41) | 0.23 (0.12, 0.41) |
| Copy | 0.94 (0.9, 0.96) | 0.23 (0.12, 0.41) | 0.27 (0.14, 0.44) | 0.23 (0.12, 0.41) | 0.27 (0.14, 0.44) |
| Count | 0.88 (0.83, 0.91) | 0.60 (0.42, 0.75) | 0.40 (0.24, 0.58) | 0.13 (0.05, 0.3) | 0.17 (0.07, 0.34) |
| Extend To Boundary | 0.93 (0.89, 0.96) | 0.77 (0.59, 0.88) | 0.47 (0.3, 0.64) | 0.07 (0.2, 0.21) | 0.1 (0.03, 0.26) |
| Extract Objects | 0.86 (0.81, 0.9) | 0.43 (0.27, 0.61) | 0.43 (0.27, 0.61) | 0.03 (0.00, 0. 17) | 0.07 (0.02, 0.21) |
| Filled and Not Filled | 0.96 (0.93, 0.98) | 0.73 (0.56, 0.86) | 0.43 (0.27, 0.61) | 0.17 (0.07, 0.34) | 0.27 (0.14, 0.44) |
| Horizontal and Vertical | 0.91 (0.91, 0.94) | 0.43 (0.27, 0.61) | 0.10 (0.03, 0.26) | 0.27 (0.14, 0.44) | 0.33 (0.19, 0.51) |
| Inside and Outside | 0.91 (0.91, 0.94) | 0.57 (0.39, 0.73) | 0.10 (0.03, 0.26) | 0.10 (0.03, 0.26) | 0.17 (0.07, 0.34) |
| Move To Boundary | 0.91 (0.91, 0.94) | 0.37 (0.22, 0.54) | 0.30 (0.17, 0.48) | 0.20 (0.1, 0.37) | 0.20 (0.1, 0.37) |
| Order | 0.83 (0.78, 0.87) | 0.27 (0.14, 0.44) | 0.23 (0.12, 0.41) | 0.27 (0.14, 0.44) | 0.27 (0.14, 0. 44) |
| Same and Different | 0.88 (0.83, 0.91) | 0.53 (0.36, 0.7) | 0.17 (0.07, 0.33) | 0.17 (0.07, 0.34) | 0.27 (0.14, 0. 44) |
| Top and Bottom 2D | 0.95 (0.91, 0.97) | 0.60 (0.42, 0.75) | 0.57 (0.39, 0.73) | 0.23 (0.12, 0.41) | 0.37 (0.21, 0.54) |
| Top and Bottom 3D | 0.93 (0.89, 0.96) | 0.50 (0.33, 0.67) | 0.03 (0.00, 0.17) | 0.20 (0.1, 0.37) | 0.27 (0.14, 0. 44) |

