# OpenReview forum: "The ConceptARC Benchmark: Evaluating Understanding and Generalization in the ARC Domain"
_TMLR — Accepted by TMLR_

### Review · Reviewer_v6Vo · 2023-05-14

**Summary Of Contributions:**

This work proposes a new benchmark in the style of Chollet’s Abstraction and Reasoning Corpus (ARC) that differs in two ways. First, rather than presenting all tasks in one large collection, the new tasks are organized into 16 concept groups. Second, the tasks are designed to be easier than those in the original ARC, since the latter may be too difficult to measure AI progress in the near term. Human solvers and three machine solvers are evaluated. The machine solvers include the first and second place winners of the ARC Kaggle competition and GPT-4. Humans substantially outperform all machine solvers, the first place winner outperforms the second place winner, and GPT-4 performs worst overall.


**Audience:**

Yes

**Claims And Evidence:**

Yes

**Requested Changes:**

Overall, I think the paper is already in very good shape. Here are a few small things:

* The related works that develop easier ARC tasks (Kim et al. 2022 in particular) are not mentioned until Section 9. These works seem extremely related, but since this is TMLR, novelty doesn’t matter. Nonetheless, it would be good to mention these works in the introduction and/or where the motivation to simplify ARC is first mentioned.
* Can you clarify how 3 different guesses per test input are collected from GPT-4? Since the temperature is set to be 0, I would have thought that the 3 guesses are often identical. I also would expect that a higher temperature might be better, though I do see that the Webb et al. work also used a temperature of 0.
* In the limitations section, it says: ““In addition to these limitations, our studies revealed that there are a small number of tasks in the ConceptARC corpus that are ambiguous—that is, for which test inputs have more than one reasonable solution.” Can you elaborate? Which tasks are these, and how did you make this determination?

### Minor
* Typo: “that are vary in complexity”
* “Pretrained” vs. “pre-trained” both used
* Typo: “each of these concepts is central to in one or more tasks”
* “Psiturk” vs. “psiTurk” both used

**Strengths And Weaknesses:**

## Strengths
* The paper is very well written. The prose is clear and persuasive. The organization is good. The motivation, experiments, and results are well-explained.
* I particularly appreciated the comparison between human and machine errors in Section 8.2.
* I also appreciated the thorough discussion of limitations of the work in Section 8.3.
* The experimental design seems sound. I did not find any issues related to the experiments or results.
* The connections to related work are also good.

## Weaknesses
* The paper suggests that all of the tasks in a concept group are related by a shared concept. There are two possible interpretations of this, each with potential issues. (1) Tasks within a concept group are designed to evaluate _only_ that concept. (2) Each task within a group involves _many_ concepts, but the _intersection_ of concepts within a group has just one shared concept.
   * The issue with (1) is that I don’t think it is actually possible to evaluate just a single concept. For example, the tasks in the Sameness group shown in Figure 2 also involve objects, shapes, and colors. One could say that objects, shapes, and colors are not “concepts”, but that would require a careful (and probably contentious) argument.
   * The issue with (2) is that we might as well take the original ARC tasks and organize them by concept, rather than making new tasks.
* The notion of a “concept” is not well-defined in this work. The “core knowledge” that ARC uses is backed by cognitive science research. Is there any criteria that one could use to determine whether something is _not_ a “concept” for the purposes of ConceptARC? Nonetheless, I am willing to look past this issue because there is a long line of work on concept learning in AI, and it is always hard to precisely define the meaning of “concept”.
* I was also sometimes confused about the word “generalization” as used in this work. For example: “the much lower accuracies of programs we tested indicates a lack of ability to generalize.” Usually, generalization means either (1) a model trained on A performs well on B; or (2) a finding that we observed in setting A can also be observed in setting B. When generalization is mentioned in this work, which kind is it, and what are A and B?
* Relatedly, I think there is a missed opportunity to examine within- and between- concept group performance, for both humans and machines. In the absence of further results, it would be good to hint at possibilities in this direction.
* The performance of GPT-4 is said to be “impressive given that it was not designed or trained for such tasks.” It is difficult to know how impressed we should be though, given that it’s the lowest performing method, and this is a new benchmark. It would be great if it were possible to design some simple baseline approach to give further context for these results. Admittedly, I can’t think of a baseline that is simple enough to easily understand but not so simple that it gets zero on all tasks.
* I found it strange that each task has three test inputs, but human participants were each only given one of the three. Why is this better than just having one test input, like most of the original ARC tasks do?

---

> ### Author Response · Authors · 2023-07-02
> **Response to Reviewer v6Vo**
>
> We would like to thank Reviewer v6Vo for their time reviewing our manuscript and for their helpful suggestions. Below, we discuss each of the points raised.
>
> > The paper suggests that all of the tasks in a concept group are related by a shared concept. There are two possible interpretations of this, each with potential issues. (1) Tasks within a concept group are designed to evaluate only that concept. (2) Each task within a group involves many concepts, but the intersection of concepts within a group has just one shared concept. And ... The issue with (2) is that we might as well take the original ARC tasks and organize them by concept, rather than making new tasks.
>
> We agree with Reviewer v6vo that it is almost impossible to construct tasks that involve one concept and once concept only, as some concepts (e.g. shape and color) tend to be involved in all problems.  Our interpretation is in between scenarios (1) and (2) above.
>
> We don't necessarily claim that the intersection of concepts within a group involves only one shared concept, but rather that in each problem, the target concept plays the most prominent role. Original ARC problems were constructed to be a challenging combination of a number of concepts (i.e. finding the right way to use each of the involved concepts is approximately equally difficult). In contrast, we aimed to create problems in which the use of "supporting" concepts is trivial, and in which only one of the involved concepts might present a challenge.  We believe we succeeded at this -- we asked the participants in our human study for written descriptions of their solution to each task, and in most cases their descriptions indeed focused on the concept we had in mind for the task. We hope to quantify this linguistic focus in future work.
>
> Organizing ARC tasks by concept indeed presents a potential alternative to our approach. However, we don't believe it would achieve the goals of ConceptARC, since many if not most of the original tasks intentionally focus on multiple concepts (this was a point explicitly made by Chollet). In the extreme, a model that only understands one concept would perform at zero on a dataset that tests at least two concepts in each problem, thus not allowing to identify its (only) area of competence. By creating tasks that focus on one primary concept, we hope that our dataset will be more precise in identifying areas of model (in)competence.
>
> > The notion of a “concept” is not well-defined in this work. The “core knowledge” that ARC uses is backed by cognitive science research. Is there any criteria that one could use to determine whether something is not a “concept” for the purposes of ConceptARC? Nonetheless, I am willing to look past this issue because there is a long line of work on concept learning in AI, and it is always hard to precisely define the meaning of “concept”.
>
> We agree that in general, it is extremely difficult to define the notion of concept, and draw an unambiguous line between "concepts" and "non-concepts". Even within well-researched cognitive science frameworks like the core knowledge theories, there does not seem to be a full agreement on the exact and exhaustive set of concepts, and oftentimes various definitions already subtly rely on pre-existing commonsense intuitions that most humans share.
>
> That being said, in creating our dataset, we largely relied on identifying intuitive aspects of core knowledge ---which we call "concepts"---already involved in the original ARC dataset, and aimed to create problems that would more precisely focus on those concepts. In other words, we believe that the set of our concepts in our study is a subset of those tested in the original ARC. This way, we hope that our problems should be at least as well justified as those in the original ARC dataset, while being more precise about what core knowledge the problems are testing.
>
> Importantly, we do not claim that the list of concepts that we tested is exhaustive. Unfortunately, is not entirely clear whether it will ever be possible to construct such an exhaustive list of concepts. On the bright side, the existence of concepts not yet covered by our problems gives the option to expand our dataset in a non-incremental manner (adding whole new concepts, rather than adding problems within existing concepts).
>
> We have added a short discussion of this issue in the paper, as follows: ''It is important to note that `"concept'" does not have a rigorous definition in cognitive science.  Here we use the term informally to refer to a facet of core knowledge that plays important roles in the original ARC tasks, and that plays more obviously central roles in the ConceptARC tasks in a given concept group.''
>
> Response continues in next comment.

---

> > ### Author Response · Authors · 2023-07-02
> > **Continuation of Response to Reviewer v6Vo**
> >
> > >I was also sometimes confused about the word “generalization” as used in this work. For example: “the much lower accuracies of programs we tested indicates a lack of ability to generalize.” Usually, generalization means either (1) a model trained on A performs well on B; or (2) a finding that we observed in setting A can also be observed in setting B. When generalization is mentioned in this work, which kind is it, and what are A and B?
> >
> > By "generalization" we mean #2 above -- an agent that can solve a task that involves a particular concept can also solve other tasks that involve that concept. We have added a clarification of this in the paper, as follows: “The generally high accuracies of humans on each concept indicates successful generalization over the different variations in each given concept group.  In contrast, the much lower accuracies of programs we tested indicates a lack of ability to generalize over the variations in a concept group, and thus a failure to develop the abstractions that ARC is meant to test.”
> >
> > > Relatedly, I think there is a missed opportunity to examine within- and between- concept group performance, for both humans and machines. In the absence of further results, it would be good to hint at possibilities in this direction.
> >
> > >The performance of GPT-4 is said to be “impressive given that it was not designed or trained for such tasks.” It is difficult to know how impressed we should be though, given that it’s the lowest performing method, and this is a new benchmark. It would be great if it were possible to design some simple baseline approach to give further context for these results. Admittedly, I can’t think of a baseline that is simple enough to easily understand but not so simple that it gets zero on all tasks.
> >
> > Perhaps we have used the word “impressive” in a slightly colloquial sense, but we do believe that any above-zero performance requires highly nontrivial zero-shot problem-solving capacity. We certainly know that we did not make our problems trivial: the winning Kaggle algorithms show far from perfect performance on our data. In other words, our data is simpler, but comparable in difficulty to the original ARC challenge. And in that original challenge, out of more than 900 participants (teams), only 36 successfully solved more than 3\% of the problems.  We have revised the relevant sentence in our paper as follows: “In addition, we showed that GPT-4's performance, while impressive given that in many cases it exceeded the second-place ARC Kaggle program even though it was not designed for or trained on these tasks, is dramatically below that of humans, which contrasts with the results of Webb et al. in testing GPT-4 on other idealized analogy domains.”
> >
> > We agree with this comment. Initially, such analysis was among the things we planned to do. Unfortunately, for humans, the performance was generally so uniformly high across most concepts that there was little room to explore the differences in within- and between-concept performance in detail.
> >
> > We ran a simple $\chi^2$ - based contingency table statistical analysis to see whether there was evidence for differences in performance between concepts. For humans, as well as for Kaggle 1st and 2nd place algorithms, performance across concepts was highly significantly non-uniform (p < 0.001). At the same time, for GPT-4, the data does not provide sufficient evidence to reject the null hypothesis that the observed differences in performance across concepts was due to chance. This negative result should be interpreted with caution, since in the relatively low-accuracy regime for all concepts, the statistical power of this test is limited. We added this analysis to the main paper (section 8.1, last paragraph).
> >
> > Response continues in next comment.

---

> > > ### Author Response · Authors · 2023-07-02
> > > **Continuation of Response to Reviewer v6Vo**
> > >
> > > >I found it strange that each task has three test inputs, but human participants were each only given one of the three. Why is this better than just having one test input, like most of the original ARC tasks do?
> > >
> > > Similar to our general goals for this project, we aimed to reduce the noise in algorithm evaluation. By creating three test inputs for each problem, we aimed to eliminate potential cases where a problem is solved correctly but for the wrong reasons. A simple example could be when the problem output is potentially “guessable”. For example, if a problem requires extracting one of 5 figures according to some rule, an algorithm that extracts a random figure might luck out on one test input, but that is exponentially less likely to happen on all three.
> > >
> > > Each human was only given one of these test inputs to limit learning/interference effects. That is, when AI algorithms are evaluated, they independently solve each test input for each problem. By only giving humans one of the possible three test inputs, we tried to keep human and machine performance results comparable. If we were to give each human participant three test tasks for each problem, we'd likely get interference between different test inputs, as well as reduced thinking/answering time for each subsequent test input. The comparison between humans and machines on such test answers would have been less fair.   We have added clarifying text on this to the paper as follows: “Each of our tasks has three different test inputs.  The purpose of creating multiple test inputs for a given task was to control for possible “shortcuts” that might enable a solver to solve a particular test input without correctly grasping the underlying rule, whereas the purpose of creating multiple tasks within a concept group was to ensure that a solver is able to generalize across different instantiations of the concept.”
> > >
> > >
> > > >The related works that develop easier ARC tasks (Kim et al. 2022 in particular) are not mentioned until Section 9. These works seem extremely related, but since this is TMLR, novelty doesn’t matter. Nonetheless, it would be good to mention these works the introduction and/or where the motivation to simplify ARC is first mentioned.
> > >
> > > We don't agree that these works are "extremely related.  While they focus on creating easier ARC tasks, they don't include the main points of our paper: the creation of tasks that systematically focus on core concepts, and the evaluation of humans, ARC solving-programs, and GPT-4 on these tasks.  We believe it is sufficient to cover these other approaches in the Related Work section.
> > >
> > >  >Can you clarify how 3 different guesses per test input are collected from GPT-4? Since the temperature is set to be 0, I would have thought that the 3 guesses are often identical. I also would expect that a higher temperature might be better, though I do see that the Webb et al. work also used a temperature of 0.
> > >
> > > GPT-4 is deterministic at temperature 0, so we only recorded one response.  We have now added results from running GPT-4 with temperature 0.5, and repeating each task prompt three times.  As shown in Table 1 of the revised paper, the higher temperature and three guesses improves GPT-4's performance on some concepts, but only by a small amount.
> > >
> > > >In the limitations section, it says: “In addition to these limitations, our studies revealed that there are a small number of tasks in the ConceptARC corpus that are ambiguous—that is, for which test inputs have more than one reasonable solution.” Can you elaborate?
> > >
> > > In the revised paper, we have revised the discussion of ambiguous tasks:
> > > "One additional limitation—for both ConceptARC and the original ARC dataset—is that there might be more than one reasonable solution for a given test input.  Like Chollet in the original ARC set, we tried to design tasks that have only one clear solution for each test input.  However, there is always the possibility of other solutions that people would find reasonable, but that would be counted as incorrect in our study.    By closely examining incorrect solutions submitted by humans in our studies, we found six (out of 480 total) test inputs that we considered to be ambiguous in this way.  There might be additional such test inputs that we did not identify. Indeed, a small number of ambiguous test inputs are likely inevitable in any corpus.  However, the effects of such ambiguity on our results are mitigated, for both humans and machines, by (1) allowing three solutions to be submitted for each test input; (2) having multiple test inputs per task; and (3) having multiple tasks per concept group.   These factors help make the aggregate robust in spite of an inevitable small subset of ambiguous test inputs."
> > >
> > > >Typo: “that are vary in complexity”
> > >
> > > >“Pretrained” vs. “pre-trained” both used
> > >
> > > >Typo: “each of these concepts is central to in one or more tasks”
> > >
> > > >“Psiturk” vs. “psiTurk” both used
> > >
> > > We have fixed these errors in the revised paper.

---

> > > > ### Comment · Reviewer_v6Vo · 2023-07-04
> > > > **Thanks for the response**
> > > >
> > > > Thank you for the very thorough and thoughtful response to my comments, and thanks especially for the additional experimental results, including the GPT-4 temperature 0.5 ones. I don't have any major outstanding concerns -- I think the paper is in good shape.

---

### Review · Reviewer_yaNs · 2023-06-03

**Summary Of Contributions:**

This work proposes ConceptARC, an extension to the Abstraction and Reasoning Corpus (ARC) benchmark, that is intended both to be more reliably solvable by humans, and to systematically explore the role of specific concepts. The study also presents results on this new benchmark for human participants, GPT-4, and two heuristic program designed to solve ARC problems. The basic finding is that human participants reliably perform well, significantly outperforming both the heuristic programs and GPT-4.







**Audience:**

Yes

**Broader Impact Concerns:**

I do not believe there are any ethical implications that would necessitate such a statement.

**Claims And Evidence:**

Yes

**Requested Changes:**

I do not have any requested changes, but the authors may wish to address some of the concerns detailed above.

**Strengths And Weaknesses:**

This work is a valuable contribution that will be very useful in the continued effort to develop AI systems with human-like abstract reasoning capabilities. The proposed benchmark makes some important improvements relative to the original ARC benchmark, and the human behavioral experiment and evaluation of GPT-4 / heuristic programs are thorough and well documented.

I have a number of comments (detailed below), but I think the paper is already in good shape, so these are really just suggestions (some of which I imagine won't be feasible to address in this particular study).

#### Comments:

- It would be helpful to add a bit more discussion concerning which features of the conceptARC problems make them more difficult for GPT-4 than the problems in Webb et al. (2022). There are a few places in the paper where the problems from Webb et al. are described as more 'idealized' but there is not much discussion otherwise. One essential difference seems to be that the conceptARC problems require object segmentation from pixel-level inputs, whereas the problems in Webb et al. (e.g. digit matrices) only involve analogies applied to individual tokens. This also relates to the first element of core knowledge listed on pg. 3, 'objectness'.
- The evaluation of error types (e.g. near misses vs. failing to grasp the concept altogether) was somewhat unsystematic. First, one of the descriptions of a common human error type involved 'giving up', which doesn't necessarily sound like a careless mistake -- it seems like this could just as easily result from a failure to grasp the underlying concept. Second, there is no description of how errors were categorized, and no quantification of the frequency of different error types. But this does seem like a potentially important aspect of the evaluation, especially if it reveals a systematic difference between humans vs. programs. One possible way to improve this section is to have blind raters judge whether errors are more likely the result of careless mistakes vs. failure to grasp the underlying concept.
- In Webb et al., analogy problems were presented using line breaks ('\n') to indicate the spatial layout of a grid, whereas it appears that GPT-4 was evaluated on conceptARC without this formatting. This seems like it could potentially affect performance. Insofar as LLMs have exposure to 2D pseudo-visual arrays, it seems likely that it often involves explicit line breaks of this sort.
- Relatedly, I wonder how human participants would perform if given the problems in the same format as GPT-4. It does seem that it might be difficult in some cases to identify objects when the problems are formatted in this way.
- Were human participants given feedback when they got problems correct? If so, an interesting comparison would be to see how GPT-4 performs when given similar feedback. In principle it could benefit from in-context learning based on this feedback.
- It would be helpful to include supplementary figures depicting some of the 'minimal' problems used to determine subject exclusion in the human study. There is a concern that these are not sufficiently minimally difficult to be treated as mere attention checks.
- It would be informative to include some measure of error for the performance estimates (in both humans and programs). Binomial confidence intervals seem like a natural choice. It might also be helpful to plot the results, either instead of or in addition to providing a table.

---

> ### Author Response · Authors · 2023-07-02
> **Response to Reviewer yaNs**
>
> We would like to thank Reviewer yaNs for their time reviewing our manuscript and for their helpful suggestions. Below, we discuss each of the points raised.
>
> >It would be helpful to add a bit more discussion concerning which features of the conceptARC problems make them more difficult for GPT-4 than the problems in Webb et al. (2022).
>
> Thank you for this suggestion! We expanded the second paragraph in section 8.1 to address this comment. (We don’t have space in this response to cut and paste in this paragraph, but please look at the revised manuscript.)
>
> In the future, it might be interesting to create a “feature-extracted” version of ConceptARC, to better understand the reason for performance differences between ConceptARC and grid-based tasks in Webb et al. Specifically, to check what proportion of this difference is due to the difficulties in finding the right representation/features, as opposed to the differences in operating on these features afterward.
>
> >The evaluation of error types (e.g. near misses vs. failing to grasp the concept altogether) was somewhat unsystematic. First, one of the descriptions of a common human error type involved ‘giving up', which doesn't necessarily sound like a careless mistake -- it seems like this could just as easily result from a failure to grasp the underlying concept. Second, there is no description of how errors were categorized, and no quantification of the frequency of different error types. But this does seem like a potentially important aspect of the evaluation, especially if it reveals a systematic difference between humans vs. programs.
>
> We agree with this comment. For now, we used manual inspection to gain simple intuitions about what those errors are. We agree that a more thorough look into the matter is warranted, and we hope to include such analysis in our future work.  We added some text about this to the Future Work section: "In the future we also plan to extend the ConceptARC benchmark to encompass additional tasks and concept groups, and to further evaluate humans and machine solvers on these new tasks, as well as to more thoroughly analyze the different types of errors made by humans and machines."
>
> >In Webb et al., analogy problems were presented using line breaks (“\n”) to indicate the spatial layout of a grid, whereas it appears that GPT-4 was evaluated on conceptARC without this formatting. This seems like it could potentially affect performance
>
> We re-ran GPT-4 on our tasks, adding “\n” after each grid row in the prompt for each task.  We did not see any substantial difference in accuracies with this amended prompt.  We have put a footnote in the paper to this effect (last paragraph of Section 7).
>
> >Relatedly, I wonder how human participants would perform if given the problems in the same format as GPT-4. It does seem that it might be difficult in some cases to identify objects when the problems are formatted in this way.
>
> We believe that comparison between humans and machines is never 100% “fair”, since the representations can never be exactly equivalent, and since humans often have prior knowledge (such as, for example, associations with different colors, e.g. expecting “black” to represent background).   That being said, we don't fully agree that asking humans to solve problems in machine-formatted way is the best solution. Instead, we think the best we can do is to present problems in the most natural way to each entity tested (text-based for LLMs, visual for humans).
>
> >Were human participants given feedback when they got problems correct? If so, an interesting comparison would be to see how GPT-4 performs when given similar feedback. In principle, it could benefit from in-context learning based on this feedback.
>
> Humans indeed knew whether or not they failed a given attempt. We agree that giving interactive feedback could, in theory, improve GPT-4 performance on our data. That being said, in our procedure we stayed true to the original ARC challenge Kaggle format, to make the results comparable between GPT-4 and other algorithms. For now, we leave this more detailed look into GPT-4 performance for future work.
>
> >It would be helpful to include supplementary figures depicting some of the 'minimal' problems used to determine subject exclusion in the human study. There is a concern that these are not sufficiently minimally difficult to be treated as mere attention checks
>
> We have added an appendix with examples of minimal tasks.
>
> >It would be informative to include some measure of error for the performance estimates (in both humans and programs).
>
> We added a version of Table 2 with binomial confidence intervals to the appendix (it did not fully fit into the main paper). For now, we did not find a way to visualize our data that would improve the quality of presentation over the table-based format.

---

> > ### Comment · Reviewer_yaNs · 2023-07-04
> > **Reply**
> >
> > Thanks very much to the authors for so thoroughly addressing all of the issues raised in my initial review. I think the idea to develop a 'feature-extracted' version of ConceptARC would be a particularly interesting direction for future work. All of my initial concerns have been addressed, and I think the work is ready for publication.

---

### Review · Reviewer_5wLx · 2023-06-30

**Summary Of Contributions:**

This paper proposes an evaluation task that extends the Abstraction and Reasoning Corpus (ARC) to cover conceptual "wrappers" to the above benchmark (e.g., above, center, inside/outside). The authors then benchmark

**Audience:**

Yes

**Broader Impact Concerns:**

Can you please include a brief discussion about manipulation and deception as it pertains to high-level abstract reasoning?

**Claims And Evidence:**

Yes

**Requested Changes:**

- Sorry if I missed it but did you discuss why/how each concept group in ConceptARC was selected? It would be good to provide some methodological description for why these were chosen.
- It would good to understand how easily others can leverage this dataset, perhaps including a code snippet or release information.

**Strengths And Weaknesses:**

Strengths
- Neat and timely study that moves beyond traditional benchmarking tasks to settings that are not only more realistic but also more difficult for existing AI systems (e.g., GPT-4).
- The paper is very well-written and is effectively ready for publication as is.

Weaknesses
- It's unclear how narrow models optimized for the task itself would perform. Instead of using general purpose learners (e.g. GPT-esque systems). How would a standard supervised learning model (e.g., finetuned ResNet perhaps) perform on each of the concept groups? This would simply provide another baseline to compare against.

---

> ### Author Response · Authors · 2023-07-02
> **Response to reviewer 5wLx**
>
> We would like to thank Reviewer 5wLx for their time reviewing our manuscript and for their comments. Below, we discuss each of the points raised.
>
> >It's unclear how narrow models optimized for the task itself would perform. Instead of using general purpose learners (e.g. GPT-esque systems). How would a standard supervised learning model (e.g., finetuned ResNet perhaps) perform on each of the concept groups? This would simply provide another baseline to compare against.
>
> In Table 1, Section 6, and Section 8.1 of the paper, we give and discuss results from running the first- and second-place programs from the ARC-Kaggle competition on the tasks in our corpus. These two programs were the best-performing methods on the ARC dataset out of 913 submissions to the competition. The competition placed no restriction on methods used (and it is almost certain that standard supervised algorithms were tried as well among those 913 submissions). In other words, we found the best available model specifically optimized for the task at hand. Most likely, standard supervised learning models like ResNet will not perform well on ARC due to 1) ARC requiring a high level of abstract reasoning, which typical computer vision architectures commonly struggle with 2) a relatively low amount of data, which makes it hard to apply typical general-purpose deep neural networks to the task.
>
> >Sorry if I missed it but did you discuss why/how each concept group in ConceptARC was selected? It would be good to provide some methodological description for why these were chosen.
>
> We chose concepts that are central in one or more tasks in Chollet’s published ARC “training” and “evaluation” sets (though Chollet's sets were not organized around specific concepts). To clarify how it was done, we changed the following paragraph at the beginning of Section 3:
>
> Original: "We chose 16 concepts, listed in the left column of \autoref{TestInputAccuracyTable}. Each of these concepts was central in one or more tasks in Chollet's published ARC training and evaluation sets, though those sets were not organized around specific concepts."
>
> Changed: `"We studied all publicly available ARC tasks and manually identified 16 concepts used in them (listed in the left column of \autoref{TestInputAccuracyTable}). Each of these concepts was central in one or more tasks in Chollet's published ARC trainingand evaluation sets, though those sets were not organized around specific concepts."
>
> > It would good to understand how easily others can leverage this dataset, perhaps including a code snippet or release information.
>
> The dataset will be released along with a readme file describing the data format and Python code snippets showing how to load the data. We will give this information in the final, published version.  In the version submitted for review, the URL for the release was redacted to preserve author anonymity.
>
> >Can you please include a brief discussion about manipulation and deception as it pertains to high-level abstract reasoning?
>
> We agree that manipulation and deception can be interpreted as requiring a certain element of abstract reasoning (e.g. if one tries to manipulate another person, one might try to predict their reactions by reasoning by analogy, imagining themselves in a similar situation). Generally, however, such Theory of Mind-like reasoning belongs to a very different branch of Cognitive Science than what we focused on. Our analogy-making tasks had no social component, hence we believe that adding a discussion of manipulation and deception might not fit well into our paper and might make confuse our future readers.

---

### Decision · Action_Editors · 2023-07-28

**Recommendation:** Accept as is

**Comment:**

This submission proposes an extension to the original Abstraction and Reasoning Corpus (ARC) by incorporating concept groups and systematic human studies. All three reviewers find the submission addressing a timely topic, the studies thorough and rigorous, and the findings useful to the community.  Eventually, all reviewers recommended acceptance.  The AE agrees.

**Audience:**

Yes

**Claims And Evidence:**

Yes